# Free mobility across group boundaries promotes intergroup cooperation
Jörg Gross ✉, Martin Götz, Katharina Reher & Filippo Toscano

Group cooperation is a cornerstone of human society, enabling achievements that surpass individual capabilities. However, groups also define and restrict who benefits from cooperative actions and who does not, raising the question of how to foster cooperation across group boundaries. This study investigates the impact of voluntary mobility across group boundaries on intergroup cooperation. Participants, organized into two groups, decided whether to create benefits for themselves, group members, or everyone. In each round, they were paired with another participant and could reward the other's actions during an 'enforcement stage', allowing for indirect reciprocity. In line with our preregistered hypothesis, when participants interacted only with in-group members, indirect reciprocity enforced group cooperation, while intergroup cooperation declined. Conversely, higher intergroup cooperation emerged when participants were forced to interact solely with out-group members. Crucially, in the free-mobility treatment – where participants could choose whether to meet an in-group or an out-group member in the enforcement stage – intergroup cooperation was significantly higher than when participants were forced to interact only with in-group members, even though most participants endogenously chose to interact with in-group members. A few 'mobile individuals' were sufficient to enforce intergroup cooperation by selectively choosing out-group members, enabling indirect reciprocity to transcend group boundaries. These findings highlight the importance of free intergroup mobility for overcoming the limitations of group cooperation.

Cooperation is considered a crucial building block for human relationships and societies. Whether between two friends, within groups, or larger, multilayered collectives, working together allows the creation of joint welfare, surpassing what is possible through individual efforts alone[1–3]. Cooperation can be broadly defined as an action that is costly for the individual, while creating a benefit for another person or group[4], thereby also creating a social dilemma: While reciprocal cooperation creates the greatest benefits for everyone, individuals can be tempted to benefit from the cooperation of others without reciprocating, hence avoiding personal costs. Such behavior, called free-riding (or defection), poses a threat to cooperation[4–6] and groups must ensure that individuals do not simply benefit from public goods without contributing to it themselves. Previous research has shown that people are willing to punish free-riding or reward cooperation, even when it is personally costly for them to do so[7–14]. Such peer punishment or reward, if effective enough[9,15,16], can make free-riding less worthwhile for the individual, thereby preventing the breakdown of cooperation within groups.

While such peer enforcement can maintain cooperation within groups[17–20], an important open question is how cooperation can develop that is not confined to certain groups[21–24], but transcends group boundaries. In the face of global challenges, such as climate change, it is crucial to understand how groups can move beyond group-exclusive cooperation and create larger, group-transcending public goods[25].

Previous theoretical work at the interface between graph theory and evolutionary game theory has already highlighted the important role of social networks in the evolution of large-scale cooperation[26–29]. In so-called structured populations, the structure of the social network can play an important role in how resistant cooperative strategies are to defection. A Cluster of cooperators that cannot be invaded by free-riders can also emerge in dynamic networks (i.e., when agents can choose who to interact with; see, e.g., refs. 30,31). A simple yet ecologically plausible network topology assumes that agents are clustered in distinct groups, within which they can cooperate to create group-exclusive benefits[32–35]. Agents in such nested structures[27,32,34,36,37] may also cooperate across group boundaries to benefit everyone in the population. Such multilevel group structures can be found in organizations, societies, and transnational relations. For example, researchers may dedicate their time and energy to their own research projects (a "selfish choice"), collaborate with colleagues within their department ("group cooperation"), or work with members from other faculties or universities on joint research projects ("intergroup cooperation"). Similarly, politicians may prioritize policies that benefit only their local community,

Department of Psychology, Social and Economic Psychology, University of Zurich, Zurich, Switzerland. ✉e-mail: gross@psychologie.uzh.ch

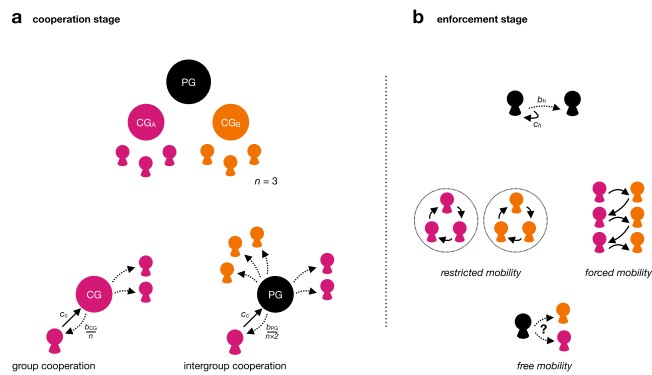

**a** cooperation stage

**b** enforcement stage

$n = 3$

CG

PG

group cooperation    intergroup cooperation

restricted mobility    forced mobility

free mobility

**Fig. 1 | Nested mobility dilemma game.** Participants were split into two groups (of size $n = 3$) and interacted over 20 rounds. Each round had two stages. In the cooperation stage (**a**), each participant received one token and simultaneously had to decide whether to keep their token, create a benefit ($b_{CG} = 1.5$) for the group ("club good" contribution [CG]; such that each group member received $1.5/3 = 0.5$) at a cost to themselves ($c_c = 1$), or create a benefit ($b_{PG} = 1.8$) for all others regardless of their group affiliation ("public good" contribution [PG]; such that everyone received $1.8/6 = 0.3$), at the same cost. In the enforcement stage (**b**), each participant in the role of the "enforcer" received an additional token and was matched with one other participant and could, conditional on the choice in the cooperation stage, decide to create a benefit $b_h = 3$ for the other participant at a cost to themselves ($c_c = 1$). Whom they were matched within this stage depended on the treatment. In the restricted-mobility treatment, enforcers only "met" other in-group members, whereas in the forced-mobility treatment, enforcers only "met" out-group members. In the free-mobility treatment, participants freely decided whether they wanted to be matched with a fellow in-group member, or an out-group member.

their country as a whole, or collaborate internationally to promote policies that benefit multiple countries, such as member states of the European Union.

When people belong to different, distinct groups and have the option to cooperate only on the group level or across group boundaries, the question arises what can motivate cooperation that goes beyond group-exclusive benefits. A recent model-based study showed that peer enforcement, in principle, can lead to the emergence of intergroup, universal cooperation[38], if in-group members frequently interact with out-group members (see also refs. [39,40]). The underlying idea is quite simple: If people meet out-group members, they do not benefit from their group cooperation and are likely to not reward such behavior. Instead, people reward out-group members if their cooperation is aimed at creating benefits for everyone, regardless of their group membership. Indeed, if people frequently meet out-group members, they begin to reward and enforce cooperation that benefits everyone regardless of their group membership, leading to a decline in group cooperation and a rise in intergroup cooperation. Hence, a simple indirect reciprocity mechanism can enforce not only (group) cooperation[41]—thereby combating free-riding—but can also foster intergroup cooperation with sufficient "relational mobility" across groups. While our focus here is on the theoretical foundations of cooperation through mechanisms of reciprocity[18,41–43], a related literature in social psychology also highlights that frequent intergroup contact could reduce intergroup hostilities and promote more positive intergroup relations[44]. Returning to the example above, researchers who frequently interact across departmental boundaries may have more opportunities to reward individuals for their efforts to advance cross-disciplinary collaboration, thereby shifting incentives toward collaboration across group boundaries.

Yet, the proposed mechanism of fostering intergroup cooperation requires that people (are forced to) have frequent exchanges with out-group members (i.e., "forced mobility"). A more realistic assumption is that people have discretion regarding whom they wish to meet and interact with. Under such 'free mobility,' it is not straightforward to assume that intergroup cooperation would emerge. In fact, it has been argued that

under this more realistic scenario, indirect reciprocity will not lead to intergroup cooperation[39,40,45]. For example, people belonging to a group may prefer to interact with other in-group members rather than with out-group members[46], and selectively enforce group cooperation in such in-group interactions. This could be because they expect more help from in-group members when cooperating at the group level and perceive intergroup cooperation as riskier in this regard. Another possible explanation is that these individuals have a general tendency to trust and cooperate more with in-group members. Thus, under these assumptions, free mobility would promote group cooperation and top-down interventions that impose sufficient intergroup exchange would be required in order for people to enforce a norm of intergroup cooperation[26].

On the other hand, under free mobility, even the mere possibility of meeting an out-group member could lead to a shift towards more group-transcending cooperation, akin to the peer punishment or reward effect observed in the literature on group cooperation[47]. That is, the possibility to reward cooperation or punish defection often increases (group) cooperation[48,49], even when rarely implemented, as people anticipate possible negative consequences of free-riding (e.g., "fear" of being punished)[8,47,50–52]. Free mobility may have a similar effect: Even if people rarely choose to meet out-group members, they may, nevertheless, cooperate beyond group boundaries to pre-empt possible negative reciprocity from out-group members for exhibiting group-exclusive cooperation.

Against this background, we tested how free mobility across group boundaries influences group vs. intergroup cooperation, and to what extent people actually use opportunities to meet out-group members to enforce intergroup cooperation. To do so, we designed a stylized nested dilemma (Fig. 1)[32–34,53–55] in which participants were assigned to one of two groups and repeatedly decided to create benefits solely for themselves, for in-group members (i.e., group cooperation), or for everyone (i.e., intergroup cooperation). After this cooperation stage (Fig. 1a), each participant was paired with another participant (Fig. 1b), learned about their group membership and previous choice, and decided whether to reward them for their action (i.e., the enforcement stage).

In two control treatments, we forced participants to always interact with either in-group members (i.e., restricted-mobility treatment), or out-group members (i.e., forced-mobility treatment) in this stage. We hypothesized that (i) less group cooperation (preregistered Hypothesis 1.1), and (ii) more intergroup cooperation (preregistered Hypothesis 1.2) emerge when individuals are 'forced' to interact across groups vs. only within groups. We further test whether this is due to a shift in how indirect reciprocity is used to enforce cooperation, such that group cooperation is rewarded more when only meeting in-group members, and intergroup cooperation is rewarded more when people are forced to only meet out-group members. In our main treatment, we allowed participants to choose whether they wanted to meet an in-group member or an out-group member in the enforcement stage (i.e., free-mobility treatment). We hypothesized that people prefer to choose and interact with in-group members rather than with out-group members (preregistered Hypothesis 2.1). Because of this, we further hypothesized that free mobility would mimic restricted mobility, with higher enforcement of group cooperation at the expense of intergroup cooperation (preregistered Hypothesis 2.2).

## Methods
### Sample
A total of 366 participants (women = 203, men = 157, non-binary = 3, undisclosed = 3; $M_{age} = 25.5$ years) participated in this study. Participants were randomly paired into groups of 6, further divided into two sub-groups of 3 (below and in the main text, a "sub-group" is called "group" for simplicity), and assigned to one of three treatments (i.e., forced mobility, restricted mobility, and free mobility; see below). As preregistered, we tested 20 groups of 6 per treatment, except for the restricted-mobility treatment, in which we tested 21 groups because of overbooking the last session.

## Nested mobility dilemma game

Participants faced a stylized social dilemma game, which we call the nested mobility dilemma game (Fig. 1). In this game, players are assigned to one of two groups (i.e., A or B). The game consists of two stages: In the cooperation stage (stage 1; Fig. 1a), each player receives one resource token. Players simultaneously decide whether to keep their tokens, invest them in their group's good, or invest them in the public good. We refer to the group's good as a club good, since in comparison to the public good, it only creates benefits for the group and excludes the other group from these benefits, whereas the public good is non-excludable[56].

Contributing the token to the club good creates a cost $c_c = 1$ for the player (i.e., the token is "spent") and a benefit $b_{CG}$ that is equally shared among the group members that the club good belongs to (i.e., "group cooperation"). Contributing the token to the public good creates the same cost $c_c = 1$ for the player (i.e., the token is "spent") and a benefit $b_{PG}$ that is equally shared among all players, regardless of their group membership (i.e., "intergroup cooperation"). To create a cooperation dilemma at both the group and the collective level we set the benefits such that $1 < b_{CG} < n_{x|y}$ and $1 < b_{PG} < n_A + n_B$. Hence, keeping the token at this stage is always more beneficial for the player, regardless of the decision of others, because $1 > b_{CG} / n_{x|y}$ and $1 > b_{PG} / (n_A + n_B)$. In other words, if others do not cooperate (regardless of whether at the group or intergroup level), it is best to also not cooperate and keep the token instead. If others cooperate (either at their own group or intergroup level), it is still best to withhold cooperation, thereby benefiting from the cooperation of others without paying the cost of cooperating oneself (i.e., free-riding). From a game-theoretic perspective, it is therefore easy to see that pure defection (i.e., all players decide to keep and do not engage in any form of cooperation) is the only pure Nash equilibrium in this game, if played finitely, since defection payoff dominates both forms of cooperation[38].

To counteract these incentives to free-ride in the cooperation stage, the game has a second stage, the "enforcement stage" (Fig. 1b). In this stage, each player is paired with another player (i.e., their "receiver") in the role of the "enforcer." The enforcer learns about the stage 1 choice, group membership, and the previous stage 2 choice of their receiver and can decide whether to reward her receiver. Rewarding creates a benefit $b_h$ for the receiver and a cost $c_h$ for the enforcer, with $0 < c_h < b_h$. Depending on the strategies of the players and payoffs, receiving a reward can compensate for the cost of cooperation, thereby promoting cooperation[38]. However, when assuming that $b_{PG}/(2n) < b_{CG}/n$ (as implemented in the experiment, see below), it is reasonable to assume that agents are more likely to reward group cooperation when meeting in-group members (because the personal benefit from group cooperation of another player is higher than from intergroup cooperation), whereas agents are more likely to reward intergroup cooperation when meeting out-group members (because they do not benefit from group cooperation in the opposing group, but do benefit from intergroup cooperation).

## Implementation

In the experiment, the participants were randomly assigned to one of two groups that remained fixed throughout the study. Each group had three members. Participants played the nested mobility dilemma game for 20 consecutive rounds. For stage 1, we set $b_{CG}$ to 1.5 and $b_{PG}$ to 1.8. We chose these parameters so that, for the individual, it is better if a person from their own group cooperates at the group level rather than at the intergroup level ($b_{CG}/n = 0.5 > b_{PG}/(2n) = 0.3$). Yet, full intergroup cooperation leads to an overall generated benefit of $0.3 \times 6 \times 6 = 10.8$ tokens whereas full group cooperation leads to an overall generated benefit of $0.5 \times 3 \times 6 = 9$ tokens. In each round, participants first decided whether to "keep" their token, "invest it into their group pool" (i.e., group cooperation option), or "invest it into the universal pool" (i.e., intergroup cooperation option). Once everyone entered their decision, they received feedback on how many tokens were invested into their group pool and the intergroup pool in total, respectively, how much they earned, and how much the other group members and the

members of the other group earned in this stage on average. This concluded the cooperation stage.

In the enforcement stage, participants received one additional token and were assigned to another participant (their 'receiver'). Here, they learned about how their receiver spent their token in the preceding cooperation stage, and whether their receiver belonged to their group or the other group. From round 2 onwards, they also learned about the enforcement choice of their receiver in the previous round. The latter was implemented to provide second-order reputation information, such that participants could withhold help not only to non-cooperative receivers but also to first-stage cooperators who, however, were not willing to reward others in the second stage (so-called second-order free-riders). Results reported in the Supplementary Information indicate that participants indeed frequently withheld help from others who did not help in the previous round (see Supplementary Note 5).

Based on this information, the participants had to decide whether to transfer their token to their receiver (i.e., help/reward) or keep the token for themselves in stage 2 (i.e., not help). In the former case, the participant lost their additional stage 2 token ($c_h = 1$) and created a benefit of 3 tokens ($b_h$) for their receiver. In the case of keeping, the receiver did not receive anything, and the participant kept their token, added to their round earnings. The round concluded with feedback on whether their enforcer decided to transfer their token to them (i.e., reward them or not) and total round earnings.

## Experimental manipulation

Across treatments, we manipulated whom participants were paired with in the enforcement stage (i.e., stage 2; Fig. 1b). In the restricted-mobility treatment, participants were paired only with other in-group members during the 20 rounds. In the forced-mobility treatment, participants were only paired with out-group members during the 20 rounds. In the free-mobility treatment, after the feedback for the cooperation stage, participants had to indicate whether they want to "meet a member of the own group" or "meet a member of the other group." Hence, in every round, each participant could freely decide whether they wanted to be paired with an in-group or an out-group member in the "enforcer" role.

Across all treatments, pairings were constructed such that two participants could not be each other's enforcers at the same time. Hence, if participant $x$ was the enforcer for participant $y$, participant $y$ would not be the enforcer for participant $x$ in that round. Furthermore, individual participants were not labeled; therefore, it was not possible to identify or keep track of individual participants across rounds. This was done to exclude direct reciprocity or reputation building across rounds such that enforcement and its effect on cooperation can only be explained through indirect reciprocity.

## Additional measures

After our main task, we measured identification with the own group (of 3), and the larger collective (of 6; item 1: "I felt a bond with my group/all participants," item 2: "I am glad that I was part of my group/the larger collective," item 3: "I felt solidarity with my group/all participants," item 4: "I felt committed to my group/all participants") on a four-point rating scale ranging from "not at all" to "very strongly"[57].

Subsequently, participants completed the incentivized six-item social value orientation slider measure[58]. In this task, participants had to make six decisions on how to allocate points between themselves and an unknown person. Points can be allocated self-servingly or pro-socially (sacrificing points to benefit the other person), allowing to estimate the degree of other-regarding concerns (i.e., social preferences). Furthermore, we probed different motives underlying cooperation and the decision to meet in-group vs. out-group members in the free-mobility treatment (see Supplementary Tables 17, 18 for additional results), measured social dominance orientation[59] and the big five personality traits[60], and finally asked for demographic information.

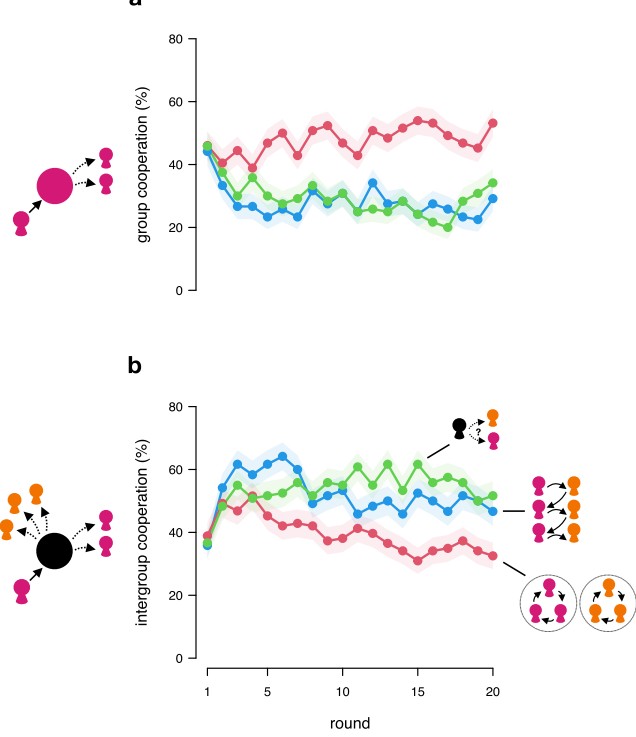

**Fig. 2 | Free and forced mobility increase intergroup cooperation.** Group cooperation (**a**) and intergroup cooperation (**b**) across rounds, measured as the average percentage of choosing to invest the token in the group's club good or public good, respectively, in the restricted-mobility treatment (red), in which participants only interact with in-group members in the enforcement stage (stage 2); the forced-mobility treatment (blue), in which participants only interact with out-group members in stage 2; and the free-mobility treatment (green), in which participants can choose whom to get paired with in stage 2. Error bands indicate the standard errors of the round means based on $n = 120$ observations per treatment and round.

## Procedures and payments

This study received ethical approval from the University of Zurich (Approval no. 22.10.5) and did not involve deception. Participants were invited to a large laboratory with separate cubicles preventing them from seeing whom they interacted with. They first received and had to agree to an informed consent to participate. All instructions were presented on the computer screen and used neutral language throughout, avoiding suggestive terms like "game," "players," "cooperation," "helping," etc. to avoid any framing effects. After the instructions, the participants had to answer 14 comprehension questions to ensure that they understood the rules of the game. In the case of questions, participants were instructed to raise their hands so that one of the two experimenters could come to their cubicle to answer any questions. The sessions lasted between 50 and 90 min. Each participant received a debriefing on the study procedures and goals upon completion. After each group in every session finished, the experimenters called the participants one by one to give them performance-based payments.

Participants received a flat fee of 10 CHF for participation. Furthermore, for each participant, one round of the nested mobility dilemma game was randomly selected by a computer for payment. One token was worth 2.60 CHF. In addition, participants could earn up to 4.30 CHF in the social value orientation slider measure. On average, the participants earned 23.20 CHF in total for their participation.

## Pre-registration

Sample size, procedures, measures, and hypotheses were preregistered on AsPredicted: https://aspredicted.org/DZX_9GJ (04/17/2023). None of the participants were excluded from the analyses, and all measurements were

preregistered and reported above. Data, materials, and analyses can be accessed at: https://osf.io/j634g/ (see also Supplementary Data 1 for the main dataset).

## Reporting summary

Further information on research design is available in the Nature Portfolio Reporting Summary linked to this article.

## Results

### Intergroup mobility fosters intergroup cooperation

As predicted, forcing people to interact across group boundaries decreased group-exclusive cooperation (Fig. 2a). Specifically, group cooperation was lower under forced mobility compared to when mobility was restricted to in-group members (multilevel logistic regression; forced vs. restricted mobility = 0.788, $p = 0.030$, 95% confidence intervals = [0.077,1.498]; Table S1), and decreased over rounds (multilevel logistic regression; round [forced mobility] = $-0.025$, $p = 0.007$; 95% confidence intervals = [$-0.0429, -0.007$]; Table S1). Analogously, forcing people to interact across group boundaries increased intergroup cooperation (Fig. 2b). While intergroup cooperation declined across rounds when people only met in-group members (multilevel logistic regression; round [restricted mobility] = $-0.058$, $p < 0.001$; 95% confidence intervals = [$-0.076, -0.040$]; Table S2), this decline was significantly attenuated when people were forced to meet only out-group members (multilevel logistic regression; round × forced mobility = 0.030, $p = 0.021$; 95% confidence intervals = [0.005,0.0563]; Table S2).

Interestingly, contrary to what we hypothesized, when people were free to choose whom to interact with, we observed cooperation patterns similar to when people were forced to meet only out-group members (Fig. 2). Specifically, group cooperation rates were low and not statistically different under free mobility from forced mobility at the average level (multilevel logistic regression; free mobility vs. forced mobility = 0.269, $p = .462$; 95% confidence intervals = [$-0.450,0.988$]; Table S1) and over rounds (multilevel logistic regression; free mobility × round vs. forced mobility = $-0.013$, $p = 0.315$; 95% confidence intervals = [$-0.038,0.012$]; Table S1). Further, group cooperation declined over rounds (compared to restricted mobility; multilevel logistic regression; free mobility × round = $-0.063$, $p < 0.001$, 95% confidence intervals = [$-0.087, -0.039$]; Table S1). Analogously, intergroup cooperation rates increased over rounds under free mobility in general (multilevel logistic regression; round [free mobility] = 0.030, $p = 0.001$, 95% confidence intervals = [0.012,0.047]; Table S2), and compared to the restricted-mobility treatment (multilevel logistic regression; free mobility × round = 0.088, $p < 0.001$, 95% confidence intervals = [0.062,0.113]).

Thus, forcing people or allowing them to choose to interact with outgroup members increased intergroup cooperation and reduced group cooperation, whereas forcing people to only interact with in-group members led to increased group cooperation at the expense of group-transcending, intergroup cooperation. After the nested mobility dilemma, participants in the forced- and free-mobility treatments also self-reported to identify less with their in-group members in favor of the larger collective that included the out-group members compared to the restricted-mobility treatment (see Supplementary Note 4, Supplementary Fig. 4, and Supplementary Tables 5–7 in the Supplementary Information for more details).

### How intergroup mobility affects the enforcement of cooperation

To understand why intergroup cooperation was more prevalent under free and forced mobility, we analyzed how indirect reciprocity was used to enforce cooperation, by looking at the stage 1 choices that were rewarded across treatments in the enforcement stage. Across all treatments, participants had 84% lower odds of rewarding free-riders than cooperators (either at the group or intergroup level; multilevel logistic regression; free-riding = $-1.860$, $p < 0.001$, 95% confidence intervals = [$-2.318, -1.401$]; Table S8 model 3). In other words, participants were more likely to reward cooperation than free-riding in all treatments.

Yet, what type of cooperation was rewarded differed across the treatments. Figure 3 illustrates the degree of enforcement of intergroup rather than group cooperation. Compared to the restricted-mobility treatment, participants had 78% (multilevel logistic regression; free-mobility treatment = −1.521, $p$ = 0.007, 95% confidence intervals = [−2.636,−0.407]; Table S9) and 82% (multilevel logistic regression; forced-mobility treatment = −1.740, $p$ = 0.002, 95% confidence intervals = [−2.842;−0.637]; Table S9) lower odds of rewarding group cooperation in the free- and forced-mobility treatment, respectively. Instead, participants had 154%

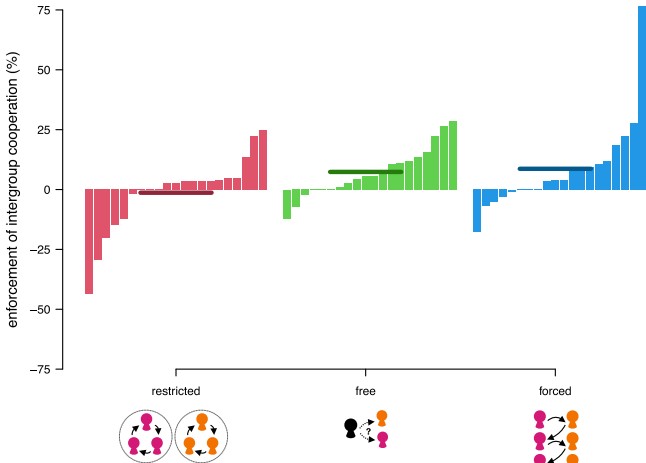

**Fig. 3 | Intergroup cooperation is enforced under both, free and forced mobility.** Each bar represents the average percentage point difference in receiving a benefit, when the person decided to cooperate on the intergroup vs. group level in stage 1 (help | [intergroup cooperation] – help | [group cooperation]) in one group. Negative values indicate that, in this group, group cooperation was rewarded more than intergroup cooperation. Positive values indicate that, in this group, intergroup cooperation was rewarded more than group cooperation. Each red bar (left) represents one group in the restricted-mobility treatment ($n$ = 21 groups). Each green bar (middle) represents one group in the free-mobility treatment ($n$ = 20 groups). Each blue bar (right) represents one group in the forced-mobility treatment ($n$ = 20 groups). The horizontal lines indicate the average differences across all groups within a treatment.

(multilevel logistic regression; free-mobility treatment × intergroup cooperation = 0.932, $p$ = 0.001, 95% confidence intervals = [0.380,1.483]; Table S9) and 147% (multilevel logistic regression; forced-mobility treatment × intergroup cooperation = 0.906, $p$ = 0.001, 95% confidence intervals = [0.362,1.450]; Table S9) higher odds of rewarding intergroup cooperation in the free- and forced-mobility treatments, respectively, compared to participants in the restricted-mobility treatment. Hence, participants who were forced to meet out-group members or who were free to choose whom to meet more strongly rewarded intergroup cooperators than group cooperators, both in general and in comparison to participants that were restricted to meeting only in-group members. For more detailed analyses on stage 2 helping choices, see Supplementary Note 5 in the Supplementary Information.

**(Not) taking advantage of free mobility**

As expected, the results revealed that forcing intergroup mobility, compared to restricting mobility to the in-group, (i) increased enforcement of intergroup cooperation, and (ii) the degree of intergroup cooperation (at the expense of group cooperation). Like forced mobility, free mobility also increased intergroup cooperation and the enforcement thereof. Regarding the cooperation and enforcement levels, choices under free mobility were remarkably similar to those under forced mobility (in contrast to Hypothesis 2.2). This may suggest that people predominantly chose to meet out-group members in the free-mobility treatment to actively enforce intergroup cooperation in the opposing group (i.e., their voluntary meeting choices mimicked the forced-mobility treatment). Contrary to this (and actually in line with Hypothesis 2.1), in only 37% of the choices, participants chose to actively meet an out-group member, 8.3% of participants ($n$ = 10) chose to never meet an out-group member (mimicking the restricted mobility experience), and only 1.7% ($n$ = 2) chose to always meet an out-group member (mimicking the forced mobility experience; see Fig. 4a).

Yet, when people decided to meet out-group members (even if they did so rarely), they strongly enforced intergroup cooperation (Fig. 4b). Specifically, when meeting an out-group member who cooperated at the group level, participants had 84% lower odds of rewarding this behavior (compared to in-group members; multilevel logistic regression; out-group member | group cooperation = −1.833, $p$ < 0.001, 95% confidence intervals = [−2.386,−1.279]; Table S10), whereas they showed a 145% increase in the odds of rewarding intergroup cooperation of out-group members

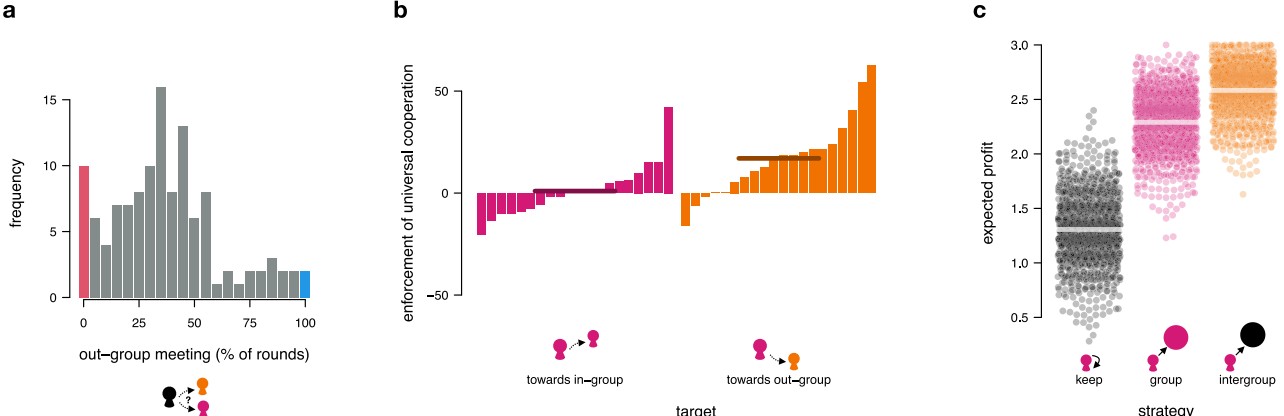

**Fig. 4 | Free mobility.** Frequency of choosing to meet an out-group member in the free-mobility treatment ($n$ = 120 participants). The red bar highlights choices perfectly mimicking restricted mobility, and the blue bar highlights choices perfectly mimicking forced mobility (**a**). Even though rare, when meeting out-group members (orange), intergroup cooperation was more strongly enforced than when meeting in-group members (pink); (**b**). Each bar represents the percentage point difference in receiving a benefit when the person decided to cooperate at the group vs. intergroup level in stage 1 (help | [intergroup cooperation] – help | [group cooperation]) in one group ($n$ = 20 groups in total), split by whether the partner was from their in- or out-group. Negative values indicate that group cooperation was rewarded more than intergroup cooperation. Positive values indicate that intergroup cooperation was rewarded more than group cooperation. Intergroup cooperation paid off (**c**). Each dot represents the expected outcome of one simulation ($n$ = 1000 simulations in total), based on the observed frequency of meeting in- vs. out-group members and their choices to reward or not in stage 2, depending on three strategies: always keeping the unit in stage 1 (free-riding; black); always cooperating at the group level (group cooperation; pink); and always cooperating on the public good level (intergroup cooperation; orange).

(compared to in-group members; multilevel logistic regression; out-group member | intergroup cooperation = 0.895, $p = 0.014$, 95% confidence intervals = [0.182,1.607]; Table S10). "Mobile" individuals who freely chose to meet out-group members also more likely rewarded intergroup cooperation, compared to individuals that were forced to meet out-group members (i.e., compared to participants in the forced-mobility treatment; multilevel logistic regression; intergroup cooperation × free-mobility treatment = 0.790, $p = 0.014$, 95% confidence intervals = [0.159,1.421]; Table S11). Supplementary Fig. 6 further shows that this higher enforcement of intergroup cooperation when meeting out-group members was rather stable across rounds.

Interestingly, even when participants chose to meet in-group members, we observed a small increase in odds (61%) to reward intergroup vs. group cooperation, which, however, was not significant at the 5% level (multilevel logistic regression; in-group member | intergroup cooperation = 0.474, $p = 0.063$, 95% confidence intervals = [−0.025,0.972]; Table S10). In comparison, the odds to reward intergroup vs. group cooperation were 1% and further decreased over rounds in the restricted-mobility treatment. More generally, across all conditions, we found that participants with higher social preferences (as measured in the social value orientation slider task) cooperated more at the intergroup level[61] (see Supplementary Note 6).

Although meeting out-group members was rare under free mobility, it paid off to cooperate across group boundaries in this treatment. To see that, we simulated the average reward in stage 2 that a person would earn by consistently following one of three different strategies across rounds: keeping the unit in stage 1 (i.e., being a free-rider), cooperating at the group level (i.e., "group cooperator"), or cooperating at the intergroup level (i.e., "intergroup cooperator"). For each simulation run, we simulated 20 rounds in which we randomly selected and recorded the helping choice of a participant that was paired with a free-rider, group cooperator, or intergroup cooperator in that round, respectively. Figure 4c shows the expected, average stage 2 earnings for each strategy under free mobility (based on 1000 simulation runs). According to the simulation results, a free-rider would earn 1 and 1.3 points less in stage 2 than a group cooperator or an intergroup cooperator, respectively. Hence, consistently keeping the unit in stage 1 (i.e., free-ride) did not pay off. Importantly, an intergroup cooperator earned 0.3 points more than a group cooperator per round (i.e., 6 points more over all 20 rounds, on average), showing that even under the low observed probability of meeting out-group members, intergroup cooperation was the most profitable strategy under free mobility.

### What motivates meeting out-group members?
Based on the choices in the cooperation stage (stage 1) and the feedback participants received before making the decision to meet an in- or out-group member, we identified two possible reasons for participants to meet out-group (rather than in-group) members. First, participants who cooperated across groups had a higher probability of choosing to interact with an out-group member in stage 2 (multilevel logistic regression; intergroup cooperation = 0.847, $p = 0.012$, 95% confidence intervals = [0.190,1.504]; Table S4).

Second, the greater the difference between out-group and in-group club good contributions, the more likely participants were to choose to meet an out-group member (multilevel logistic regression; other − own group cooperation = 0.541, $p = 0.006$, 95% confidence intervals = [0.155,0.928]; Table S4). Hence, members of groups that already cooperated across group boundaries selectively sought out-group members who exhibited more group cooperation (compared to their own group), possibly to enforce stronger intergroup cooperation in the opposing group. In addition, when asked about their motivation to meet out-group members, participants most frequently reported that they wanted to motivate members of the other group to cooperate across groups (see Supplementary Table 17 for details). It is important to note that these exploratory results provide only correlational, rather than causal, evidence for the reasons behind choosing to meet in-group versus out-group members.

## Discussion
Numerous mechanisms have been identified that groups use to discourage free-riding and uphold group cooperation[1,4,12,62]. What is often overlooked is that group cooperation requires defining who is part of the group and allowed to benefit from shared resources, and who is excluded[63]. As a result, distinct group boundaries and identities can emerge[64–67], setting the stage for intergroup conflict and the inability to address common challenges collaboratively across group boundaries[63,68].

Here, we showed that intergroup mobility, the ability to meet and selectively enforce the actions of out-group members, plays a critical role in overcoming group cooperation. When enforcement (and, thus, indirect reciprocity) is confined to in-group members, we observe the emergence of groups that predominantly cooperate within their group boundaries and self-report a stronger identification with their group (Figure S4 and Table S6; see also ref. 69). To establish cooperation beyond these limited groups, norms of intergroup cooperation must be enforced across group boundaries.

Our study demonstrates that even the occasional possibility of interacting with out-group members can be sufficient to enforce such norms. One potential explanation for the advantage of free mobility is that individuals anticipate potential negative reciprocity (i.e., withholding rewards for group cooperation) from out-group members and proactively cooperate for the collective benefit. When mobility is restricted to in-group members, people, on the other hand, only need to anticipate or react to whatever their fellow group members deem "praiseworthy." Furthermore, we showed that a minority of mobile "intergroup cooperation enforcers" exists that selectively choose to interact with out-group members and enforce intergroup cooperation in these interactions. These intergroup cooperation enforcers may play a pivotal role in making intergroup cooperation more attractive to members of other groups.

Outside our controlled laboratory environment, several factors may impede free mobility. Geographical separation, language barriers, strong in-group identification, or institutional restrictions imposed by leaders or groups may reduce or prevent interactions with out-group members, reinforcing group-exclusive cooperation. For example, countries may impose travel restrictions, constraining the mobility of their members to (systematically or unintentionally) curtail cooperation across group boundaries and foster stronger in-group identification. Yet, due to increased global trade, ease of travel, and education, individuals are arguably now more mobile than ever in human history[70]. Previous research has indeed demonstrated that heightened relational mobility[71] and identification as "global citizens" can lead to increased cooperation beyond specific group affiliations[35,72,73] (see also ref. 74 for similar determinants of cross-group cooperation in bonobos).

## Limitations
Our results specify a mechanism through which (the possibility of) cross-group mobility can foster intergroup cooperation. While our stylized experiment can provide causal evidence of how the reach of indirect reciprocity can enforce group or intergroup cooperation, it should be noted that social structures and interaction patterns are much more complex outside of the laboratory. People often belong to multiple larger or smaller groups and share different identities[56,65], and the meeting frequency with members of other groups may dynamically change, governed by higher-order social institutions (such as travel restrictions) or social norms (such as implicitly shared rules of not interacting with strangers). Moreover, larger group sizes might decrease the probability of being paired with an out-group member and rewarded for intergroup cooperation. Future research is needed to determine whether the proposed mechanism holds in larger groups (for theoretical results on group size in fixed and forced mobility, see also ref. 38). Furthermore, we did not investigate the nested dilemma without a reward stage. From a theoretical perspective, without any enforcement, we should expect a decline of any cooperation, similar to the decline in cooperation usually observed in linear public goods games (see, e.g., refs. 8,9,75). Previous empirical work on the nested dilemma, indeed, observed declining

cooperation patterns[56], whereas[37] observed rather stable group cooperation (and a declining trend of intergroup cooperation). With our data, however, we cannot address how the enforcement stage shifted the cooperation patterns compared to a situation in which rewards were not possible. For example, it is conceivable that, without reward, groups already exhibit a tendency to cooperate across group boundaries, in which case restricting rewards to the in-group reduces intergroup cooperation. Hence, our study can only provide insight into how the restriction of the scope of rewards (in stage 2) can shift cooperation (in stage 1).

It is important to note that we chose a relatively high cost-benefit ratio of rewarding in line with previous studies[10,76] and as often used in the peer punishment literature[8,9,15]. Yet, this cost-benefit ratio may have motivated a high rate of rewarding (see also Supplementary Fig. 3, 5), that to some extent may have been motivated by efficiency concerns[77] over and beyond purely rewarding certain forms of cooperation. Additionally, the choice to use reward rather than punishment was motivated by previous research on the helping game and indirect reciprocity[41,43,78]. Yet, previous research on the public goods game also frequently used punishment, instead[7–9,11]. Future research could investigate whether intergroup cooperation can be likewise reinforced by peer punishment instead of rewards, and whether lower cost-benefit ratios would still be enough to foster cooperation in stage 1. Finally, we found evidence for our preregistered hypotheses that people prefer to choose and interact with in-group members rather than with out-group members in the free-mobility treatment. While we provided some correlational evidence for the reasons behind meeting out-group members, we can only speculate why people, in the majority of rounds, prefer to be paired with an in-group member. From a purely strategic perspective, people may expect more help from in-group members, particularly when they cooperate at the group level, or they would like to be paired with in-group members to favor them financially rather than providing additional resources to out-group members. Future work is needed to delve deeper into the motivations of free meeting choices, possibly also providing novel insights into mechanisms that motivate higher intergroup mobility.

## Conclusion

Mobility allows a central mechanism for curtailing free-riding—indirect reciprocity[41,42,79–82]—not only to influence group members, but also to establish public goods from which everyone benefits, regardless of their group membership. This mechanism could also explain how smaller groups can merge into larger collectives and how group-defining characteristics (i.e., group-exclusive club goods and local "cultural identities") can disappear in favor of large-scale cooperation. Especially in times when humanity faces collective challenges, such as climate change, necessitating collective action across group boundaries, intergroup mobility may play a pivotal role in enabling groups to move beyond the limited scope of group cooperation and foster intergroup cooperation.

## Data availability

The data and analysis code are deposited at: https://osf.io/j634g (https://doi.org/10.17605/OSF.IO/J634G).

## Code availability

All analyses code and code for running the experiment (programmed in HTML/PHP and JavaScript) are deposited at: https://osf.io/j634g (https://doi.org/10.17605/OSF.IO/J634G).

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

## Acknowledgements

We would like to thank Timon Annen, Alexandra Gartmann, Noémie Lushaj, and Eliane Marti for their help in collecting the data. This work has received funding from the Swiss State Secretariat for Education, Research and Innovation (SERI) to J.G. The funders had no role in study design, data collection and analysis, decision to publish or preparation of the manuscript.

## Author contributions

J.G. conceived the research, J.G., M.G., K.R., and F.T. designed the research, J.G. conducted the study and analyzed data, J.G. wrote the manuscript with critical revisions from M.G., K.R., and F.T.

## Competing interests

The authors declare no competing interests.
