## [Peer Review file · Communications Psychology]

Free mobility across group boundaries promotes intergroup cooperation

Corresponding Author: Professor Jörg Gross

Version 0:

Decision Letter:

Dear Professor Gross,

Thank you for your patience during the peer-review process. Your manuscript titled "Free mobility across group boundaries promotes intergroup cooperation" has now been seen by 3 reviewers, and I include their comments at the end of this message. They find your work of interest but raised some important points. We are interested in the possibility of publishing your study in Communications Psychology, but would like to consider your responses to these concerns and assess a revised manuscript before we make a final decision on publication.

We therefore invite you to revise and resubmit your manuscript, along with a point-by-point response to the reviewers. Please highlight all changes in the manuscript text file.

Editorially, we consider that you should justify your design choices, elaborate further on the study's motivation, and provide a deeper interpretation of the results, ideally incorporating more theoretical perspectives. Another important concern is the transparency and reproducibility of the data. R1 suggested sharing the data, and R2 emphasized the importance of clear data presentation. We request you share your data in an online repository and provide the reviewers with an anonymous link to the storage location. Additionally, R3 offered suggestions for improving the data analysis, such as examining the temporal dynamics of intergroup reward processes and second-order reputation effects. Please either perform these additional analyses or provide a clear justification for not doing so.

I am attaching an Editorial Requests Table that details critical reporting requirements for the revised manuscript. Please attend to each item and ensure your manuscript is fully compliant. We are requesting that your manuscript aligns with these requirements as this facilitates the evaluation of your manuscript, reducing delays in re-review and potential future acceptance. If your revised manuscript is not aligned with these requests on major issues, such as those concerning statistics, it may be returned to you for further revisions without re-review. Additional information can be found in our style and formatting guide Communications Psychology formatting guide.

Please use the following link to submit your

- revised manuscript,
- point-by-point response to the referees' comments,
- cover letter (as a separate document),
- the Editorial Policy Checklist (see below),
- the Reporting Summary (see below), and
- the completed Editorial Request Table (attached):

Link Redacted

Best regards,

Yafeng Pan

Yafeng Pan, PhD
Editorial Board Member
Communications Psychology
orcid.org/0000-0002-5633-8313

REVIEWER EXPERTISE:

Reviewer #1: group cooperation/relational mobility

Reviewer #2: group cooperation

Reviewer #3: group cooperation

REVIEWER REPORTS:

Reviewer #1 (Remarks to the Author):

This study examines how voluntary movement between groups affects cooperation among those groups. In a lab experiment, participants were divided into two groups and had to decide whether to create benefits for themselves, their group, or everyone. They were paired with another participant each round and could reward the other's actions in an 'enforcement stage,' which allowed for indirect reciprocity.

When participants only interacted with their own group, cooperation within the group was strong, but cooperation between groups was weaker. However, when participants were forced to interact only with members of the other group, cooperation between groups improved.

In a key part of the experiment, participants were allowed to choose whom to interact with—either someone from their group or from the other group. Despite most choosing to interact within their group, intergroup cooperation remained high. A few participants who chose to interact with the other group helped to promote cooperation across group boundaries, showing that mobility is essential for broader cooperation.

I have enjoyed reading this paper. I find it quite comprehensive and clearly written, and introducing novel and important experimental results that will surely also inspire future research along these lines. For these reasons, I am in favor of publication subject to the following revisions.

As the authors may be aware, research at the interface of evolutionary game theory and network science has played an important role in elucidating how various types of groups interactions and mobility influence the evolution of cooperation. In this regard the introduction should be further improve and note also theoretical research along these lines. Relevant reviews from a while ago are: Evolutionary dynamics of group interactions on structured populations: A review, *J. R. Soc. Interface* 10, 20120997 (2013) and Coevolutionary games - A mini review, *BioSystems* 99, 109-125 (2010). More recent reviews are Reputation and reciprocity, *Phys. Life Rev.* 46, 8-45 (2023) and Statistical physics of human cooperation, *Phys. Rep.* 687, 1-51 (2017). I hope at least some of these reviews will be useful to the authors as they explain why the results are as they are from the theoretical point of view.

Also, it would be useful if the authors would make their data from the experiments available as supplementary material. This would promote the usage of this research and allow also others to take better advantage of this research, and also allow them to reproduce the results.

If a revision is granted, I will be happy to review the manuscript again.

Reviewer #2 (Remarks to the Author):

See attachment.

Reviewer #3 (Remarks to the Author):

This paper investigates the impact of intergroup mobility on intergroup cooperation. The experiment involved participants playing a public goods game, with both local and global components, followed by an enforcement stage where participants could reward another participant. The reward recipient could be an in-group member, an out-group member, or selected by the participant. The experiment was pre-registered and the authors followed it closely. The results indicate that intergroup mobility supports sustained intergroup cooperation, whether this mobility is free or imposed. Overall, I found the paper to be well-written, and the results align with the authors' claims. However, I have a few comments and suggestions for improvement.

- My first comment is about the dynamic of intergroup rewarding. As this is the main reason why intergroup cooperation are high in the free mobility treatment, I would like to see the dynamics of intergroup encounters. Does intergroup rewarding occur consistently from the first round, or does it gradually increase as the rounds progress? Providing this temporal insight would help clarify the mechanisms behind sustained intergroup cooperation.

- My second comment is about the size of the groups. In the experiment the groups are relatively small (3 members and 6 members in total). We can expect that as the size of the groups increases, the probability of being rewarded for intergroup cooperation in the free mobility condition would likely decrease. This could, in turn, reduce the overall level of intergroup cooperation as group size increases. I would encourage the authors to discuss this potential limitation, as it might mitigate the conclusion about the effect of intergroup mobility on cooperation at larger scale.

- Finally, I was wondering what would be the effect of the second-order reputation information. According to the methods section, participants were aware of their recipient's previous enforcement choices, which should logically influence their decision to reward or not. I don't see this variable included in the regression analysis (Table S4-S7). Incorporating this variable into the analysis could be interesting.

Minor comments:

- In Figure 2 the colored bands around the mean are not described in the legend.

- In the restricted treatment, the level of intergroup cooperation is not null (Fig. 3). Do the individuals who engage in intergroup cooperation in this condition share any characteristics with those who enforce cooperation in the free mobility condition?

- I was wondering why the authors did not cite nor use the supplementary figures in the results section? Including references to these figures where relevant could strengthen some of the paper's key arguments and provide additional support for the findings.

EDITORIAL POLICIES

We ask that you ensure your manuscript complies with our editorial policies and reporting requirements.

To that end, we require revised manuscripts to be accompanied by two completed items: a reporting summary that collects information on study design and procedure, and an editorial policy checklist that verifies compliance with all required editorial policies.

- <https://www.nature.com/documents/nr-reporting-summary.zip>>Nature Research Reporting Summary
- <https://www.nature.com/documents/nr-editorial-policy-checklist.pdf>>Editorial Policy Checklist

All points on the policy checklist must be addressed. Your revised manuscript can only be sent back to the referees if these checklists are completed and uploaded with the revision.

Notes: If you have submitted a Stage 1 Registered Report, Review, Primer, Comment, or Perspective you do not need to submit these forms. If you have already submitted these forms, you may disregard this request.

** Visit Nature Research's author and referees' website at  for information about policies, services and author benefits**

Version 1:

Decision Letter:

Dear Professor Gross,

Your manuscript titled "Free mobility across group boundaries promotes intergroup cooperation" has now been seen by our reviewers, whose comments appear below. In light of their advice I am delighted to say that we are happy, in principle, to publish a suitably revised version in Communications Psychology.

We therefore invite you to revise your paper one last time to address the remaining concerns of our reviewers and a list of editorial requests. At the same time we ask that you edit your manuscript to comply with our format requirements and to maximise the accessibility and therefore the impact of your work.

EDITORIAL REQUESTS:

SUBMISSION INFORMATION:

OPEN ACCESS:

Link Redacted

We hope to hear from you within four weeks; please let us know if you need more time.

Best regards,

Jennifer Bellingtier

Jennifer Bellingtier, PhD
Senior Editor
Communications Psychology

Yafeng Pan, PhD
Editorial Board Member
Communications Psychology
orcid.org/0000-0002-5633-8313

REVIEWER EXPERTISE:

Reviewer #1: group cooperation/relational mobility

Reviewer #2: group cooperation

Reviewer #3: group cooperation

REVIEWERS' COMMENTS:

Reviewer #1 (Remarks to the Author):

The authors have revised their manuscript comprehensively and with love to detail. I warmly recommend publication in present form.

Reviewer #2 (Remarks to the Author):

The authors took my previous comments seriously and addressed them in a satisfactory way. I appreciate the efforts and I have no further points to raise.

Reviewer #3 (Remarks to the Author):

I appreciate the authors' efforts in addressing the concerns I raised. I have no further inquiries.

REVIEWER REPORTS

Reviewer #1

This study examines how voluntary movement between groups affects cooperation among those groups. In a lab experiment, participants were divided into two groups and had to decide whether to create benefits for themselves, their group, or everyone. They were paired with another participant each round and could reward the other's actions in an 'enforcement stage,' which allowed for indirect reciprocity.

When participants only interacted with their own group, cooperation within the group was strong, but cooperation between groups was weaker. However, when participants were forced to interact only with members of the other group, cooperation between groups improved.

In a key part of the experiment, participants were allowed to choose whom to interact with—either someone from their group or from the other group. Despite most choosing to interact within their group, intergroup cooperation remained high. A few participants who chose to interact with the other group helped to promote cooperation across group boundaries, showing that mobility is essential for broader cooperation.

I have enjoyed reading this paper. I find it quite comprehensive and clearly written, and introducing novel and important experimental results that will surely also inspire future research along these lines. For these reasons, I am in favor of publication subject to the following revisions.

Authors' Response

Thank you very much for your kind words and the time and effort you put into reviewing our manuscript; we greatly appreciate it. Below, you can find a detailed point-by-point response to the points you raised.

As the authors may be aware, research at the interface of evolutionary game theory and network science has played an important role in elucidating how various types of groups interactions and mobility influence the evolution of cooperation. In this regard the introduction should be further improve and note also theoretical research along these lines. Relevant reviews from a while ago are: Evolutionary dynamics of group interactions on structured populations: A review, *J. R. Soc. Interface* 10, 20120997 (2013) and Coevolutionary games - A mini review, *BioSystems* 99, 109-125 (2010). More recent reviews are Reputation and reciprocity, *Phys. Life Rev.* 46, 8-45 (2023) and Statistical physics of human cooperation, *Phys. Rep.* 687, 1-51 (2017). I hope at least some of these reviews will be useful to the authors as they explain why the results are as they are from the theoretical point of view.

Authors' Response

Thank you very much for pointing us to these fundamental reviews of reciprocity, network structures, and human cooperation. In this paper, we, of course, use a very simplified 'network structure' in which participants are clustered in two distinct groups and can only provide benefits to their group or the overall collective (i.e., also a somewhat restricted 'strategy space'). Based on our understanding, the theoretical literature on evolutionary game theory on graphs goes far beyond that, considering a large variety of interaction

structures and dynamic networks. In our revision, we integrated this work by referring to the excellent reviews you pointed to, so that readers are aware of this rich theoretical literature. In particular, in the Introduction (p. 2-3), we draw some analogies and write:

“Previous theoretical work at the interface between graph theory and evolutionary game theory has already highlighted the important role of social networks in the evolution of large-scale cooperation^{26–29}. In so-called structured populations, the structure of the social network can play an important role in how resistant cooperative strategies are to defection. Cluster of cooperators that cannot be invaded by free-riders can also emerge in dynamic networks (i.e., when agents can choose who to interact with; see, e.g., ^{30,31}). A simple yet ecologically plausible network topology assumes that agents are clustered in distinct groups, within which they can cooperate to create group-exclusive benefits^{32–35}. Agents in such nested structures^{27,32,34,36,37} may also cooperate across group boundaries to benefit everyone in the population. Such multilevel group structures can be found in organizations, societies, and transnational relations [...]”

Our main goal with these revisions was to (briefly) draw a connection and create awareness of this work, which, while particularly important, also considers much more complex social network structures. If the Reviewer thinks that we should draw more attention to this work (or more specific work on nested dilemma structures, like, for example, Wang et al., 2011¹, which we now cite), we would be happy to include it, of course.

Also, it would be useful if the authors would make their data from the experiments available as supplementary material. This would promote the usage of this research and allow also others to take better advantage of this research, and also allow them to reproduce the results.

Authors' Response

Thank you for encouraging us to adhere to open science practices. We gladly shared our data and materials for future use. To facilitate this, we had already posted all (i.e., raw and processed) data, analysis code, and code to run the experiment on the open science framework (OSF) prior to our first submission. The link to the repository can be found in the Data and Code Availability statements of the manuscript (https://osf.io/j634g/?view_only=606440f2eb334c0a90e19da4f22d17b8).

In addition, we now submitted the data file directly as Supplementary Material as part of our revision, as you suggest. If the Editor agreed, readers interested in the data would then have direct access from the journal website, and could find all other materials (e.g., analysis script, code to run the study, etc.) from the OSF repository.

Reviewer #2

The authors report on an original experiment on cooperation within and between groups. Participants are organized in “sessions” of 6, which are split in groups of 3. The experiment is repeated for 20 rounds. Each round consists of 2 steps. In the first step, participants are endowed with a token and must decide whether to keep it for themselves, contribute it to the “group” public good, or contribute it to the “global” public good. Contributions to the public good provide benefit to the ingroup, while contributions to the global good provide benefit for both the ingroup and outgroup. Both types of contributions are individually costly but socially beneficial, making the overall game a nested social dilemma. In the second step, participants are matched in pairs, are informed of the decision of their match in the 1st step and must decide whether to reward the match or not. Rewarding comes at a cost for the decision maker, but is efficiency-enhancing because rewards received by the target are much greater than the cost.

There are 3 conditions, which differ only by the matching process in the second step. In one condition, the matching in the second step is strictly within group (baseline). In another condition, matching is strictly between group (Forced mobility), and in the last conditions, participants choose who to interact with.

The main result is that both forced and free mobility increase intergroup cooperation, leading to greater overall efficiency. Interestingly, the authors show that free mobility is effective in sustaining cooperation despite not being used, i.e. participants generally choose to be matched with an ingroup.

The paper provides an interesting contribution to the literature on (nested) social dilemma, and how cooperation can be sustained in such setting. The design is clean, with mostly pre-registered hypotheses, which makes the principal results both credible and easily interpretable. The writing is efficient and clear. It was mostly an interesting and engaging read.

Authors' Response

We would like to thank you very much for your constructive feedback and, especially, your willingness and time to review the manuscript; we greatly appreciate it. Below, you can find a detailed point-by-point response to all of the points you raised.

I have the following comments on the design, motivation and data analysis that should be considered before publication:

Major comments:

I have 3 major comments that concern the main message of the paper:

1. The motivation is exposed somewhat quickly, and the link between the design and real word situations is not clearly explained. It would help the reader if the authors would elaborate on these points a little bit. What type of situations do the authors have in mind? What are examples of situations in which one can choose to contribute to a local vs a global good, and can (choose) to reward ingroups or outgroups? As it stands, I am not totally sure about the type of situation (or theory) that this experiment informs.

Authors' Response

Thank you for your comment. Our research is, indeed, motivated by more theoretical literature on reciprocity in nested dilemmas^{2,3}. Indirect reciprocity has been shown to be an important mechanism to foster group cooperation. Yet, under specific group structures (like in the nested dilemma) and from a theoretical perspective, indirect reciprocity may be unable to sustain intergroup cooperation but rather enforce group-exclusive cooperation. We interpret our empirical results as somewhat more optimistic in this regard, as even under 'free mobility,' while people tend to not 'take full advantage' of this mobility, we still observe high levels of intergroup cooperation.

Yet, to put the nested dilemma and dyadic helping more into context, we now illustrate (a) nested dilemma structures with more concrete examples and (b) also provide an example for intergroup exchanges along the lines of our (rather stylized but therefore very controlled) experiment on p. 2-3 and p. 4:

"A simple yet ecologically plausible network topology assumes that agents are clustered in distinct groups, within which they can cooperate to create group-exclusive benefits³²⁻³⁵. Agents in such nested structures^{27,32,34,36,37} may also cooperate across group boundaries to benefit everyone in the population. Such multilevel group structures can be found in organizations, societies, and transnational relations. For example, researchers may dedicate their time and energy to their own research projects (a 'selfish choice'), collaborate with colleagues within their department ('group cooperation'), or work with members from other faculties or universities on joint research projects ('intergroup cooperation'). Similarly, politicians may prioritize policies that benefit only their local community, their country as a whole, or collaborate internationally to promote policies that benefit multiple countries, such as member states of the European Union. When people belong to different, distinct groups and have the option to cooperate only on the group level or across group boundaries, the question arises what can motivate cooperation that goes beyond group-exclusive benefits."

"While our focus here is on the theoretical foundations of cooperation through mechanisms of reciprocity^{18,41-43}, a related literature in social psychology also highlights that frequent intergroup contact could reduce intergroup hostilities and promote more positive intergroup relations⁴⁴. Returning to the example above, researchers who frequently interact across departmental boundaries may have more opportunities to reward individuals for their efforts to advance cross-disciplinary collaboration, thereby shifting incentives toward collaboration across group boundaries."

Moreover, we now discuss the limitations of this approach in the Discussion (p. 22-23):

"Our results specify a mechanism through which (the possibility of) cross-group mobility can foster intergroup cooperation. While our stylized experiment can provide causal evidence of how the reach of indirect reciprocity can enforce group or intergroup cooperation, it should be noted that social structures and interaction patterns are much more complex outside of the laboratory. People often belong to multiple larger or smaller groups and share different identities^{56,64}, and the meeting frequency with members of other groups may dynamically change, governed by higher-order social institutions (such as travel restrictions) or social norms (such as implicitly shared rules of not interacting with strangers)."

2. The effect of the reward stage is not clear. How does the existence of a reward stage impact selfish choices, local contributions and global contributions? To illustrate why it matters: take the decisions from the no mobility baseline. How would it compare to a baseline without the reward stage? Several hypotheses are possible: it might reduce selfish choices at the benefit of (ingroup) cooperation. But it could also reduce global cooperation at the benefit of ingroup cooperation. If the second hypothesis holds true, then the effect of the reward stage on efficiency / cooperation might be negative. This can severely impact the interpretation of the results. For instance, one could argue that ingroup rewards is what generates parochialism in cooperation. In this case, reward to the ingroup reduces global cooperation, and reward to the outgroup may merely reestablish it.

Authors' Response

This is, of course, a very valid point. We did not perform a control condition consisting only of the nested dilemma stage without the reward stage. Hence, we have no empirical benchmark to compare the results of the other control conditions with. Instead, we can (only) observe and document the relative shift in cooperation when mobility is in-group restricted, out-group restricted, or free (i.e., the main goal of our study).

From a theoretical perspective, we simply did not consider this treatment since we would expect a convergence to full defection when rewarding is not possible due to the setup of the payoffs; Regardless of group or intergroup cooperation, free-riding is always advantageous for individuals. We describe this logic (and the motivation to add a reward stage for that reason) in the Methods section in the second and third paragraph of the “nested mobility dilemma game” part (p. 7-8). This also fits empirical results from repeated public goods games (without punishment or reward) that usually observe a decline in cooperation across rounds, due to these advantages of free-riding.

To the best of our knowledge, only a few studies reported data from a ‘baseline’ repeated nested dilemma (without punishment or other types of institutions that change the equilibrium of the game). Chakravarty and Fonseca⁴ found the typical decline of cooperation for both group and intergroup cooperation, as one would also expect for the outlined theoretical reasons and data from repeated linear public goods games. Otten et al.⁵, on the other hand, observed rather stable group cooperation. In unpublished data from our own lab, albeit from a different project and slightly different setup, we also observed declining intergroup, yet somewhat more stable, group cooperation in the nested dilemma.

While we tentatively would expect either a decline in cooperation, generally, or more (and more stable) group cooperation without any reward opportunities, we agree with the Reviewer that it ultimately remains an open question of how reward shifts cooperation compared to a no-reward baseline treatment.

We now discuss this important point now in the (newly added) Limitations section of the Discussion (p. 23):

“Furthermore, we did not investigate the nested dilemma without a reward stage. From a theoretical perspective, without any enforcement, we should expect a decline of any cooperation, similar to the decline in cooperation usually observed in linear public goods game (see, e.g., ^{8,9,74}). Previous empirical work on the nested dilemma, indeed, observed declining cooperation patterns⁵⁶, whereas ³⁷ observed rather stable group cooperation (and a declining trend of intergroup cooperation). With our data, however, we cannot address how

the enforcement stage shifted the cooperation patterns compared to a situation in which rewards were not possible. For example, it is conceivable that, without reward, groups already exhibit a tendency to cooperate across group boundaries, in which case restricting rewards to the in-group reduces intergroup cooperation. Hence, our study can only provide insight into how the restriction of the scope of rewards (in stage 2) can shift cooperation (in stage 1)."

3. Relatedly, the authors should discuss why they expected an ingroup bias in the first place, and exactly how this is supported by the data. The results can be explained without any parochialism, simply by beliefs about the reaction of others to my decisions. For instance, if I expect the others to reward me only if I am good to them (for instance by contributing to the ingroup when the sender in the reward phase is from the ingroup), then I could decide to contribute more to the "local" good than to the "global" group even without any attachment to the group. This mechanism would be neutralized in a baseline without the reward phase. The data on group identification in Figure S1 is not very transparent, since we do not now the raw levels of ingroup / outgroup identification, only the gap between the two. It is therefore not extremely convincing.

(Note that I don't believe that collecting new data is a condition for publication, but points 2 & 3 should be considered, and explanation should be provided).

Authors' Response

We (hope to correctly) assume the Reviewer here refers to Hypothesis 2.1 (that people prefer to choose and interact with in-group members rather than with out-group members in stage 2 when given the possibility) when referring to "ingroup bias."

In this regard, we completely agree that cooperation choices (when forced to get paired with in-group members), as well as meeting choices (in the free-mobility treatment) may be strategically motivated, driven by beliefs about the likelihood of being rewarded. Indeed, we cannot say too much about the motivations – whether there is a somewhat 'irrational' / 'purely psychologically' preference for preferring to get paired with in-group members (which could be called parochialism) or whether this is simply driven by beliefs about how others reward behavior. For this we reason, we also refrain from using the term parochialism in the manuscript.

The identification questions, to our mind, also do not resolve this issues but rather add an interesting different perspective. Because identification was measured after the experiment, it may inform the development of 'group identity' based on experiences made in the experiment. For a 'purely psychological' parochialism effect, we would need to ask these questions prior to any interactions to check whether people already have a favorable view of their in-group from the get-go (and why this is).

To address this, we now discuss the potential underlying motivations in the Introduction (p. 4) and the newly added Limitations section in the Discussion (p. 24):

"For example, people belonging to a group may prefer to interact with other in-group members rather than with out-group members⁴⁶, and selectively enforce group cooperation in such in-group interactions. This could be because they expect more help from in-group members when cooperating at the group level and perceive intergroup cooperation as riskier in this regard. Another possible explanation is that these individuals have a general tendency to trust and cooperate more with in-group members. Thus, under these assumptions, free

mobility would promote group cooperation and top-down interventions that impose sufficient intergroup exchange would be required in order for people to enforce a norm of intergroup cooperation²⁶.

“Finally, we found evidence for our pre-registered hypotheses that people prefer to choose and interact with in-group members rather than with out-group members in the free-mobility treatment. While we provided some correlational evidence for the reasons behind meeting out-group members, we can only speculate why people, in the majority of rounds, prefer to be paired with an in-group member. From a purely strategic perspective, people may expect more helping from in-group members, particularly when they cooperate at the group level, or they would like to be paired with in-group members to favor them financially rather than providing additional resources to out-group members. Future work is needed to delve deeper into the motivations of free meeting choices, possibly also providing novel insights into mechanisms that motivate higher intergroup mobility.”

Furthermore, we adapted Figure S1 (now Figure S4) to also show levels of in-group vs. collective identification rather than only the comparison of differences and added regression results (Tables S5–S7) on all three dependent variables (i.e., in-group identification, ‘global’ identification, and the difference).

Other points:

I have several other points that concern secondary results from the paper:

1. What is the difference of reward behavior when one meets an outgroup vs an ingroup? Does this depend on the cooperation decision? Does it depend on the treatment? (e.g. reward to an ingroup when chosen and when forced). I might have missed it, but are ingroup rewarded more for ingroup or outgroup cooperation?

Authors’ Response

We added a new Figure (Figure S5) to the Supplementary Information that illustrates the full reward contingencies (i.e., reward frequency depending on treatment × group membership × cooperation choice of partner) and added new sections and analyses to the Supplementary Information (see section “detailed results on helping choices”) to provide more information on the helping patterns.

Table S9 shows that group cooperation was rewarded less in the forced- (and free-) meeting treatment (compared to the restricted, in-group meeting treatment), whereas intergroup cooperation was rewarded more in the forced- (and free-) meeting treatment (compared to the restricted meeting treatment). Table S11 further shows that out-group members were helped more when they exhibited intergroup cooperation (compared to group cooperation) in the forced-mobility treatment. This effect was even stronger in the free-mobility treatment, in which meeting an out-group member was a deliberate choice (i.e., partner cooperated across groups × free interaction).

Disentangling (motives behind) reward choices is challenging and our experiment was rather designed to see how a change in the rules of stage 2 influences stage 1 (which has exactly the same contingencies and consequences across all treatments). But we hope the new analyses more transparently present this data now and also point to some interesting (exploratory) findings that could also inform future research.

2. Some design choices are not well justified. For instance, why choose reward over punishment as a cooperation enforcement device? Why make the reward system so cost-effective / efficiency- improving?

Authors' Response

Indeed, the decision to choose reward was mainly because our underlying model was informed and inspired by the indirect reciprocity literature (that often uses the helping game as a reward mechanism) than the punishment literature. Also, we drew some conceptual analogies to 'intergroup contact,' which may be more based on rewarding certain behavior rather than punishing it (especially when meeting out-group members). However, these are no strong theoretical reasons, and we now acknowledge this in the Discussion (p. 23-24):

“Additionally, the choice to use reward rather than punishment was motivated by previous research on the helping game and indirect reciprocity^{41,43,77}. Yet, previous research on the public goods game also frequently used punishment, instead^{7-9,11}. Future research could investigate whether intergroup cooperation can be likewise reinforced by peer punishment instead of rewards, and whether lower cost-benefit ratios would still be enough to foster cooperation in stage 1.”

We further discuss the point about cost-effectiveness. This was mainly motivated by our theoretical model². Yet, in hindsight, we also think that a lower cost/benefit ratio (like 1:2) should, theoretically, also lead to stable cooperation in groups of three and would potentially have been a stronger test of the reward mechanism (i.e., also creating somewhat stronger incentives for free-riding).

Nevertheless, one reason for making rewards in this stage so cost-effective, from a purely experimental point of view, is, of course, that it gives a clear test of how the manipulation (restricted, forced, free mobility) shifts cooperation, since the cost/benefit ratio is the same across treatments.

We now reflect on these points in the Discussion and write (p. 23):

“It is important to note that we chose a relatively high cost-benefit ratio of rewarding in line with previous studies^{10,75} and as often used in the peer punishment literature^{8,9,15}. Yet, this cost-benefit ratio may have motivated high rate of rewarding (see also Supplementary Figure S3 and S5), that to some extent may have been motivated by efficiency concerns⁷⁶ over and beyond purely rewarding certain forms of cooperation.”

3. Relatedly, how can we interpret the very high level of reward? How often does one get rewarded even if selfish? Such behavior is actually likely for deciders who are concerned about efficiency.

Authors' Response

Also related to the point above (and to the newly added Figure S5 and discussion in the 'detailed results on helping choices' section); We indeed created an environment in which not getting rewarded is quite 'punishing' and the high reward rate may also be a consequence of participants not wanting to be 'too harsh' to others and forgo the opportunity to create the points for others (even if it produced a cost for them). So, the point about efficiency preferences is well-taken.

We now discuss this point also more extensively in the manuscript and Supplementary Information (see newly added section ‘detailed results on helping choices’).

We would, nevertheless, like to note that the main focus is on the differences across conditions. If behavior would only be driven by efficiency concerns (i.e., try to create the maximum for everyone), people should always cooperate towards the public good and reward, irrespective of the behavior of the receiver. Hence, while we acknowledge that efficiency preferences may play a role in the general motivation to reward, we do not fully see how this would explain the different cooperation patterns in stage 1 across treatments.

4. According to the description of the reward stage, participants were informed of the reward decision of their match in the previous round. Why did the authors choose to give this information? This information might confound the motivation to reward: do people reward cooperation or previous reward behavior? The authors should justify this choice, and probably control for this in the regressions explaining reward behavior.

Authors’ Response

Indeed, this is a crucial point. From a theoretical perspective, the game introduces a second-order free-riding problem. Since rewarding (and stage 1 cooperation) is costly, it is in one’s best interest if others enforce cooperation through costly rewards, while oneself saves the cost of rewarding. From a game-theoretic logic, rewards should not work if second-order free-riding cannot be ‘punished.’ For this reason (for a more model-based discussion, see ²), we provided information on the reward choices of receivers.

And, indeed, we now provide some empirical evidence that participants take this information into account and ‘punished’ second-order free-riders by withholding help (see also the newly added Figure S5). This, in our view, is also a crucial psychological insight: Participants not only seemed to attempt to ‘maintain’ (group or intergroup) cooperation through rewards (i.e., punish free-riders), but also maintained the functioning of the reward mechanism itself (i.e., by punishing second-order free-riders).

To us, this is not necessarily a confound but rather shows that people integrate different motives in their decision to punish, which may allow them to simultaneously (a) use rewards to gear stage 1 choices and (b) maintain the functioning of the reward mechanism itself. We now discuss this design choice in the Methods section (p. 10):

“In the enforcement stage, participants received one additional token and were assigned to another participant (their ‘receiver’). Here, they learned about how their receiver spent their token in the preceding cooperation stage, and whether their receiver belonged to their group or the other group. From round 2 onwards, they also learned about the enforcement choice of their receiver in the previous round. The latter was implemented to provide second-order reputation information, such that participants could withhold help not only to non-cooperative receivers but also to first-stage cooperators who, however, were not willing to reward others in the second stage (so-called second-order free-riders). Results reported in the Supplementary Information indicate that participants indeed frequently withheld help from others who did not help in the previous round (see the ‘detailed results on helping choices’ section in the Supplementary Information for these results).”

Moreover, we extensively revised the Supplementary Information, ran additional analyses, and modified all previous regression models to control for/include this motive (i.e., whether the target helped previously or not; see revised Table S8–S11).

5. The information that participants receive at the end of the contribution stage is not easy to interpret. While participants received detailed information about the decisions of the ingroups, they are only informed about the total earnings of the other group members (but not their contributions to the inter or intragroup public good). It seems very hard to deduce the group contribution of the members of the other group from this information. Therefore, participants have better information about the decisions of the ingroup members. This could explain why participants prefer being matched to the ingroup: with better information about the ingroup, it is easier to interpret their individual behavior (for instance relative to the average behavior in the group), and therefore make better informed rewarding decision. The choice of this information structure should be justified.

Authors' Response

We believe that this may be a misunderstanding. After stage 1, each participant learned about how much they earned, how much their own group members earned, on average, and how much members of the other group earned on average. Hence, we did not provide full feedback for every individual participant after stage 1, since we also did not want to introduce strong individual-level reputation concerns but rather wanted to give feedback on “group outcomes” (also to exclude the possibility of direct reciprocity). However, after matching, the participants received detailed feedback on the *individual* choices of the matched partner. Hence, participants, to some degree, also had the ‘choice’ to receive more detailed information about one specific individual and his/her detailed contribution choices. From this perspective, the argument could also be that if participants wanted to know more about how an out-group member decided, in detail, they should have preferred to meet an out-group member.

Still, we agree that the meeting choice may not only reflect “where to enforce” (in-group/out-group) but also “what to learn” (more detailed information about an in-group vs. out-group member; maybe also see Table S17 and S18, although we did not consider such an ‘information’ motive here, in particular).

6. A clearer exposition of (raw) data would help. For instance, how did selfishness differ across treatments? Figure S2 is not clear cut, and a brief discussion of this in the text would help. Same holds for the reward behavior.

Authors' Response

We now added an additional model on free-riding (i.e., the third dependent variable we have for stage 1). According to the results (reported in Table S3, see also Figure S1), selfishness was higher in the first round under free and forced mobility, and increased in all treatments across rounds, except significantly less so in the free mobility treatment. Here, we found a negative treatment \times round interaction. However, in general, free-riders remained in the minority.

Regarding the reward behavior, we now added extensive new analyses in the revised Supplementary Information and discuss these in the ‘detailed results on helping choices’ section. We hope that this provides a clearer and more extensive presentation of (albeit rather complex) helping choices.

7. Did you have a measure of preference for efficiency? It might be a better predictor than SVO for the choice between ingroup and intergroup cooperation.

Authors' Response

Thank you for this suggestion. Unfortunately, we did not include a measure of preference for efficiency, and the SVO measure we used⁶, as you already point out, does not provide a clear distinction between social preferences and efficiency preferences.

8. How does reward received (or not) at round t impact contribution decisions at time t+1? Reward, just like punishment, is a way to “teach” others what we expect from them.

Authors' Response

We now provide three new tables (see Table S12–S14) showing how helping shifted choices depending on whether it was rewarded in stage 2 of the previous round. This shows that receiving help generally increased the type of cooperation that was rewarded (see columns 2 and 3 / difference between ‘not helped’ and ‘helped’). For keeping, the results are more mixed. Keeping, in general, was less stable (see percentage difference) and only systematically decreased when not helped in the free-mobility treatment. One reason may be that reward (compared to punishment) is more effective in reinforcing cooperation than deterring free-riding. However, it is also important to note that keeping was rather rare and cooperation rates were rather high. Hence, we have fewer observations for the keeping observations (see lines 1 and 2 in Table S12–S14). More generally, we hope that this provides some evidence that rewarding shifted behavior in meaningful ways.

8. At the bottom of page 14, the authors make an interpretation that does not seem entirely correct. “Second, the more the opposing group cooperated within their own groups, the more likely participants chose to meet an out-group member” is supported by the positive and significant coefficient on the difference between other and own group cooperation ($b=0.541$). This latter variable also increases when own group contribution decreases, therefore the parameter $b=0.541$ is not appropriate to support their claim.

Authors' Response

Thank you for pointing this out to us. We now rephrased our reporting of this result to clarify that this conclusion is based on difference scores (p. 20-21):

“Second, the greater the difference between out-group and in-group club good contributions, the more likely participants were to choose to meet an out-group member (multilevel logistic regression; other – own group cooperation = 0.541, $p = .006$; Table S4).”

9. The phrasing at the beginning of the last paragraph of page 14 suggests a causal interpretation of what is actually a correlation: “*we identified two reasons for participants to choose meeting out- group (rather than in-group) members (suggests causality). First, participants who cooperated across groups had a higher probability of choosing to interact with an outgroup member (correlation)*” This is slightly misleading and should be edited.

Authors’ Response

We now rephrased the sentence on p. 20 to “*we identified two possible reasons for participants to meet out-group (rather than in-group) members*” and end the “What motivates meeting out-group members?” section on p. 21 with the sentence: “*It is important to note that these exploratory results provide only correlational, rather than causal, evidence for the reasons behind choosing to meet in-group versus out-group members.*”

We hope these revisions clarify that all three results are correlational by nature, as you correctly pointed out. Of course, we are happy to adapt further if you think we were still implying causality too starkly.

Finally, we would like to thank you once more for your in-depth review of our work; we consider our manuscript to have significantly improved because of it.

Reviewer #3

This paper investigates the impact of intergroup mobility on intergroup cooperation. The experiment involved participants playing a public goods game, with both local and global components, followed by an enforcement stage where participants could reward another participant. The reward recipient could be an in-group member, an out-group member, or selected by the participant. The experiment was pre-registered and the authors followed it closely. The results indicate that intergroup mobility supports sustained intergroup cooperation, whether this mobility is free or imposed. Overall, I found the paper to be well-written, and the results align with the authors' claims. However, I have a few comments and suggestions for improvement.

Authors' Response

Thank you very much for your valuable feedback and the time you dedicated to reviewing our manuscript; we greatly appreciated your input. Please find below a comprehensive response addressing each point you raised in detail.

- My first comment is about the dynamic of intergroup rewarding. As this is the main reason why intergroup cooperation are high in the free mobility treatment, I would like to see the dynamics of intergroup encounters. Does intergroup rewarding occur consistently from the first round, or does it gradually increase as the rounds progress? Providing this temporal insight would help clarify the mechanisms behind sustained intergroup cooperation.

Authors' Response

Thank you for this suggestion. We now present the temporal dynamics in more detail. Specifically, we added a section in the revised Supplementary Information ('helping dynamics in the free-mobility treatment') that shows the rewarding pattern, depending on the round, the group of the receiver (in-group or out-group), and the cooperation choice of the receiver in stage 1. The newly added Figure S6 shows a consistent difference in rewarding intergroup cooperation more than group cooperation when the target was an out-group member. This difference was not observed for in-group targets. Descriptively, it also looks like that this difference increases over rounds (particularly due to decreased helping of group cooperation; see Figure S6).

However, according to the newly fitted model reported in Table S15, we did not find statistical support for the round trend. While we cannot interpret this lack of support as support for the lack of an effect of rounds, we did observe a significant difference in rewarding intergroup cooperation more than group cooperation in round 1, suggesting that intergroup rewarding occurs somewhat consistently from the start.

- My second comment is about the size of the groups. In the experiment the groups are relatively small (3 members and 6 members in total). We can expect that as the size of the groups increases, the probability of being rewarded for intergroup cooperation in the free mobility condition would likely decrease. This could, in turn, reduce the overall level of intergroup cooperation as group size increases. I would encourage the authors to discuss this potential limitation, as it might mitigate the conclusion about the effect of intergroup mobility on cooperation at larger scale.

Authors' Response

While we do not have a strong intuition that group size would necessarily decrease rewarding rates (since group members cannot identify each other across rounds and everyone is paired with each other), we agree that, for practical reasons (i.e., prioritizing the number of groups per treatment rather than the number of individuals per group), we opted for small groups. While the findings of group size in simple (linear) public goods, to the best of our knowledge, are somewhat mixed^{7,8}, we also agree that it remains an open question as to how larger groups would influence cooperation dynamics.

We now discuss this point in the newly added Limitations section on p. 22-23:

“While our stylized experiment can provide causal evidence of how the reach of indirect reciprocity can enforce group or intergroup cooperation, it should be noted that social structures and interaction patterns are much more complex outside of the laboratory. People often belong to multiple larger or smaller groups and share different identities^{56,64}, and the meeting frequency with members of other groups may dynamically change, governed by higher-order social institutions (such as travel restrictions) or social norms (such as implicitly shared rules of not interacting with strangers). Moreover, larger group sizes might decrease the probability of being paired with an out-group member and rewarded for intergroup cooperation. Future research is needed to determine whether the proposed mechanism holds in larger groups (for theoretical results on group size in fixed and forced mobility, see also ³⁸).”

- Finally, I was wondering what would be the effect of the second-order reputation information. According to the methods section, participants were aware of their recipient's previous enforcement choices, which should logically influence their decision to reward or not. I don't see this variable included in the regression analysis (Table S4-S7). Incorporating this variable into the analysis could be interesting.

Authors' Response

This is, of course, a very valid point that was also raised by Reviewer 2. Indeed, there is statistical support that previous enforcement choices did influence helping rates. Therefore, according to the model results, people took second-order reputation into account. We now re-ran all regression models, added new regression models on helping, and restructured the Supplementary Information, adding a new part ('detailed results on helping choices') that outlines all results on stage 2 helping choices. In the new models, we included the previous helping choices ('Partner helped (t-1)') and interactions. Earlier reported effects remain, but these models now reveal significant main effects of previous helping choice and, hence, are now reported as such.

We thank you for this suggestion. It now indeed shows that people not only enforce certain forms of cooperation with stage 2 but also seem to 'punish' second-order free-riding

(thereby, potentially helping to sustain the effectiveness of stage 2 enforcement). We also discuss this point now in the Methods section and refer the reader more directly to the results in the Supplementary Information :

“From round 2 onwards, they also learned about the enforcement choice of their receiver in the previous round. The latter was implemented to provide second-order reputation information, such that participants could withhold help not only to non-cooperative receivers but also to first-stage cooperators who, however, were not willing to reward others in the second stage (so-called second-order free-riders). Results reported in the Supplementary Information indicate that participants indeed frequently withheld help from others who did not help in the previous round (see the ‘detailed results on helping choices’ section in the Supplementary Information for these results).” (p.10)

Minor comments:

- In Figure 2 the colored bands around the mean are not described in the legend.

Authors’ Response

Thank you for pointing this oversight out to us. We now added a description to the legend. We must admit that this also made us realize that the description in the Supplementary Information figures was misleading. The colored bands indicate the standard errors of the round means; we revised the figure legends accordingly.

- In the restricted treatment, the level of intergroup cooperation is not null (Fig. 3). Do the individuals who engage in intergroup cooperation in this condition share any characteristics with those who enforce cooperation in the free mobility condition?

Authors’ Response

This was a bit hidden in the Supplementary Information, as we did not show the regression results, but we indeed found that individual-level social preferences (as measured in the separate social value orientation [SVO] task) are correlated with intergroup cooperation. People who shared more money with others in the SVO task (indicating higher social preferences) tended to cooperate more at the intergroup level, and we did not observe significant differences in this relationship across treatments. We now report these results more transparently and added the regression table in the Supplementary Information (see Table S16). From this, we would conclude that individuals who engage in intergroup cooperation, across conditions, indeed share this characteristic of being, generally, more ‘pro-social.’

- I was wondering why the authors did not cite nor use the supplementary figures in the results section? Including references to these figures where relevant could strengthen some of the paper’s key arguments and provide additional support for the findings.

Authors’ Response

Indeed, our initial aim was to use the Supplementary Information to provide additional results for the interested reader but we did not readily refer to it in the original manuscript. In the revised manuscript, we now refer more clearly to the Supplementary Information. If the

Reviewer thinks that a certain result should be highlighted more in the main text, we are, of course, happy to accommodate that.

In conclusion, we would like to thank you again for your comprehensive evaluation of our research; we believe it has substantially enhanced the quality of our manuscript.

References

1. Wang, J., Wu, B., W. C. Ho, D. & Wang, L. Evolution of cooperation in multilevel public goods games with community structures. *EPL* **93**, 58001 (2011).
2. Gross, J. *et al.* The evolution of universal cooperation. *Sci. Adv.* **9**, eadd8289 (2023).
3. Schnell, E. & Muthukrishna, M. Indirect reciprocity undermines indirect reciprocity destabilizing large-scale cooperation. *Proc. Natl. Acad. Sci. U.S.A.* **121**, e2322072121 (2024).
4. Chakravarty, S. & Fonseca, M. A. Discrimination via exclusion: An experiment on group identity and club goods. *J. Public Econ. Theory* **19**, 244–263 (2017).
5. Otten, K., Buskens, V., Przepiorka, W., Cherki, B. & Israel, S. Cooperation, punishment, and group change in multilevel public goods experiments. *Eur. Econ. Rev.* **164**, 104682 (2024).
6. Murphy, R. O., Ackermann, K. A. & Handgraaf, M. J. J. Measuring social value orientation. *Judgm. Decis. Mak.* **6**, 771–781 (2011).
7. Nosenzo, D., Quercia, S. & Sefton, M. Cooperation in small groups: The effect of group size. *Exp. Econ.* **18**, 4–14 (2015).
8. Li, T. & Noussair, C. N. Conditional cooperation and group size: Experimental evidence from a public good game. *J. Econ. Sci. Assoc.* **10**, 98–112 (2024).

Report on “Free mobility across groups boundaries promotes intergroup cooperation”.

The authors report on an original experiment on cooperation within and between groups. Participants are organized in “sessions” of 6, which are split in groups of 3. The experiment is repeated for 20 rounds. Each round consists of 2 steps. In the first step, participants are endowed with a token and must decide whether to keep it for themselves, contribute it to the “group” public good, or contribute it to the “global” public good. Contributions to the public good provide benefit to the ingroup, while contributions to the global good provide benefit for both the ingroup and outgroup. Both types of contributions are individually costly but socially beneficial, making the overall game a nested social dilemma. In the second step, participants are matched in pairs, are informed of the decision of their match in the 1st step and must decide whether to reward the match or not. Rewarding comes at a cost for the decision maker, but is efficiency-enhancing because rewards received by the target are much greater than the cost.

There are 3 conditions, which differ only by the matching process in the second step. In one condition, the matching in the second step is strictly within group (baseline). In another condition, matching is strictly between group (Forced mobility), and in the last conditions, participants choose who to interact with.

The main result is that both forced and free mobility increase intergroup cooperation, leading to greater overall efficiency. Interestingly, the authors show that free mobility is effective in sustaining cooperation despite not being used, i.e. participants generally choose to be matched with an ingroup.

The paper provides an interesting contribution to the literature on (nested) social dilemma, and how cooperation can be sustained in such setting. The design is clean, with mostly pre-registered hypotheses, which makes the principal results both credible and easily interpretable. The writing is efficient and clear. It was mostly an interesting and engaging read. I have the following comments on the design, motivation and data analysis that should be considered before publication:

Major comments:

I have 3 major comments that concern the main message of the paper:

1. The motivation is exposed somewhat quickly, and the link between the design and real world situations is not clearly explained. It would help the reader if the authors would elaborate on these points a little bit. What type of situations do the authors have in mind? What are examples of situations in which one can choose to contribute to a local vs a global good, and can (choose) to reward ingroups or outgroups? As it stands, I am not totally sure about the type of situation (or theory) that this experiment informs.
2. The effect of the reward stage is not clear. How does the existence of a reward stage impact selfish choices, local contributions and global contributions? To illustrate why it matters: take the decisions from the no mobility baseline. How would it compare to a baseline without the

reward stage? Several hypotheses are possible: it might reduce selfish choices at the benefit of (ingroup) cooperation. But it could also reduce global cooperation at the benefit of ingroup cooperation. If the second hypothesis holds true, then the effect of the reward stage on efficiency / cooperation might be negative. This can severely impact the interpretation of the results. For instance, one could argue that ingroup rewards is what generates parochialism in cooperation. In this case, reward to the ingroup reduces global cooperation, and reward to the outgroup may merely reestablish it.

3. Relatedly, the authors should discuss why they expected an ingroup bias in the first place, and exactly how this is supported by the data. The results can be explained without any parochialism, simply by beliefs about the reaction of others to my decisions. For instance, if I expect the others to reward me only if I am good to them (for instance by contributing to the ingroup when the sender in the reward phase is from the ingroup), then I could decide to contribute more to the “local” good than to the “global” group even without any attachment to the group. This mechanism would be neutralized in a baseline without the reward phase. The data on group identification in Figure S1 is not very transparent, since we do not know the raw levels of ingroup / outgroup identification, only the gap between the two. It is therefore not extremely convincing.

(Note that I don't believe that collecting new data is a condition for publication, but points 2 & 3 should be considered, and explanation should be provided).

Other points:

I have several other points that concern secondary results from the paper:

1. What is the difference of reward behavior when one meets an outgroup vs an ingroup? Does this depend on the cooperation decision? Does it depend on the treatment? (e.g. reward to an ingroup when chosen and when forced). I might have missed it, but are ingroup rewarded more for ingroup or outgroup cooperation?
2. Some design choices are not well justified. For instance, why choose reward over punishment as a cooperation enforcement device? Why make the reward system so cost-effective / efficiency-improving?
3. Relatedly, how can we interpret the very high level of reward? How often does one get rewarded even if selfish? Such behavior is actually likely for deciders who are concerned about efficiency.
4. According to the description of the reward stage, participants were informed of the reward decision of their match in the previous round. Why did the authors choose to give this information? This information might confound the motivation to reward: do people reward cooperation or previous reward behavior? The authors should justify this choice, and probably control for this in the regressions explaining reward behavior.

5. The information that participants receive at the end of the contribution stage is not easy to interpret. While participants received detailed information about the decisions of the ingroups, they are only informed about the total earnings of the other group members (but not their contributions to the inter or intragroup public good). It seems very hard to deduce the group contribution of the members of the other group from this information. Therefore, participants have better information about the decisions of the ingroup members. This could explain why participants prefer being matched to the ingroup: with better information about the ingroup, it is easier to interpret their individual behavior (for instance relative to the average behavior in the group), and therefore make better informed rewarding decision. The choice of this information structure should be justified.
6. A clearer exposition of (raw) data would help. For instance, how did selfishness differ across treatments? Figure S2 is not clear cut, and a brief discussion of this in the text would help. Same holds for the reward behavior.
7. Did you have a measure of preference for efficiency? It might be a better predictor than SVO for the choice between ingroup and intergroup cooperation.
8. How does reward received (or not) at round t impact contribution decisions at time $t+1$? Reward, just like punishment, is a way to “teach” others what we expect from them.
9. At the bottom of page 14, the authors make an interpretation that does not seem entirely correct. “Second, the more the opposing group cooperated within their own groups, the more likely participants chose to meet an out-group member” is supported by the positive and significant coefficient on the difference between other and own group cooperation ($b=0.541$). This latter variable also increases when own group contribution decreases, therefore the parameter $b=0.541$ is not appropriate to support their claim.
10. The phrasing at the beginning of the last paragraph of page 14 suggests a causal interpretation of what is actually a correlation: “*we identified two reasons for participants to choose meeting out-group (rather than in-group) members (suggests causality). First, participants who cooperated across groups had a higher probability of choosing to interact with an outgroup member (correlation)*” This is slightly misleading and should be edited.